# The Structural, Electronic, Magnetic and Elastic Properties of Full-Heusler Co2CrAl and Cr2MnSb: An Ab Initio Study

**Sara J. Yahya [1], Mohammed S. Abu-Jafar [1,\*], Said Al Azar [2], Ahmad A. Mousa [3], Rabah Khenata [4], Doha Abu-Baker [1] and Mahmoud Farout [1]**

[1] Department of Physics, An-Najah National University, Nablus P.O. Box 7, Palestine
[2] Department of Physics, Faculty of Science, Zarqa University, Zarqa 13132, Jordan
[3] Department of Basic Sciences, Middle East University, Amman 11831, Jordan
[4] Laboratoire de Physique Quantique et de Modélisation Mathématique de la Matière (LPQ3M), Université de Mascara, Mascara 29000, Algeria
[\*] Correspondence: mabujafar@najah.edu

**Abstract:** In this paper, the full-potential, linearized augmented plane wave (FP-LAPW) method was employed in investigating full-Heusler $Co_2CrA1$'s structural, elastic, magnetic and electronic properties. The FP-LAPW method was employed in computing the structural parameters (bulk modulus, lattice parameters, $c/a$ and first pressure derivatives). The optimized structural parameters were determined by generalized gradient approximation (GGA) for the exchange-correlation potential, $V_{xc}$. Estimating the energy gaps for these compounds was accomplished through modified Becke–Johnson potential (mBJ). It was found that the conventional Heusler compound $Co_2CrA1$ with mBJ and CGA approaches had a half-metallic character, and its spin-down configuration had an energy gap. It was also found that the conventional and inverse Heusler $Cr_2MnSb$ and tetragonal (139) ($Co_2CrA1$, $Cr_2MnSb$) compounds with a half-metallic character had direct energy gaps in the spin-down configuration. To a certain degree, the total magnetic moments for the two compounds were compatible with the theoretical and experimental results already attained. Mechanically, we found that the conventional and inverse full-Heusler compound $Co_2CrAl$ was stable, but the inverse $Cr_2MnSb$ was unstable in the ferromagnetic state. The conventional Heusler compound $Cr_2MnSb$ was mechanically stable in the ferromagnetic state.

**Keywords:** full-Heusler compound; electronic band structure; magnetic order; elastic properties; FP-LAPW

## 1. Introduction

Since their discovery in 1903, Heusler compounds have found many applications including spintronics [1], shape-memory devices [2] and thermoelectric power generators [3]. Heusler compounds have a type of face-centered cubic (fcc) crystal structure. These compounds can be categorized into two classes: XYZ (half-Heuslers), which consist of three FCC sub-lattices, and $X_2YZ$ (full-Heuslers), which have four FCC sub-lattices, where transition elements are represented by X and Y, and the s, p … elements are represented by Z [4].

Full-Heusler $X_2YZ$ compounds crystallize in two kinds of inverse and conventional forms. Conventional Heusler compounds crystallize in a $Cu_2MnAl$ structure with a space group of Fm-3m (space group number 225) having atomic positions of $X_2$ (1/4,1/4,1/4), (3/4,3/4,3/4), Y (1/2,1/2,1/2) and Z (0,0,0). Inverse Heusler compounds crystallize in a $Hg_2CuTi$ structure with space group of F-43m (space group number 216) having atomic positions of $X_2$ (1/4,1/4,1/4), (1/2,1/2,1/2), Y (3/4,3/4,3/4) and Z (0,0,0) [5]. Some Heusler compounds have a half-metallic (HM) character [5–18], where only a single conduction spin channel exists for half metals. For one spin channel, the spin-polarized band structure

shows metallic behavior. On the other hand, at the Fermi level, the other spin band structure shows a gap. Therefore a 100% spin polarization is exhibited by half-metallic materials.

Certain studies have explored the electronic, magnetic, elastic and structural properties of these compounds using different methods. Zhang et al. [19] focused on the $Co_2CrAl$ Heusler compound's electronic band structure and transport properties. They measured the lattice parameter, magnetic moment and indirect band gap and found those to be 5.74 , 3 $\mu_B$ and 0.475 eV, respectively. Hakimi et al. [20] conducted an experimental study of the $Co_2CrAl$ compound's magnetic and structural properties. They found that the conventional $Co_2CrAl$'s total magnetic moment was 2 $\mu_B$.

Ozdogan and Galanakis [21] determined half-metallic antiferromagnetic $Cr_2MnSb$'s magnetic and electronic properties for both conventional and inverse structure types. They found that for both structural types, $Cr_2MnSb$ is a half-metallic ferrimagnetic compound for a broad array of lattice constants.

Heusler alloys are well-known for their potential application in the spin-transfer torque (STT) sector. These materials crystallize in multifaceted structures in both cubic and tetragonal symmetries with multiple magnetic sublattices. Galanakis [22] conducted research on the magnetic and electronic properties of both full-Heusler and half-Heusler alloys. The full-Heusler alloys investigations included $Co_2MnSi$ and $Co_2MnGe$, and the half-Heusler alloys included PtMnSb, CoMnSb and NiMnSb.

Atsufumi Hirohata et al. [23] reviewed the development of anti-ferromagnetic (AFM) Heusler alloys for the replacement of iridium as a critical raw material (CRMs). They established correlations between the crystalline structure of these alloys and the magnetic properties, i.e., antiferromagnetism. This study revealed that the Heusler alloys consisting of elements with moderate magnetic moments require perfectly or partially ordered crystalline structures to exhibit AFM behavior. Using elements with large magnetic moments, a fully disordered structure was found to show either AFM or ferrimagnetic (FIM) behavior. The considered alloys may become useful for device applications due to the additional increase in their anisotropy and grain volume being able to sustain AFM behavior above room temperature.

Recently, Abu Baker et al. [24] investigated the elastic, electronic, magnetic and structural characteristics of half-metallic ferromagnetic full-Heusler alloys, namely conventional $Co_2TiSn$ and inverse $Zr_2RhGa$, employing the FP-LAPW technique. The lattice parameters for the conventional $Co_2TiSn$ and inverse $Zr_2RhGa$ were found to be 6.094 $A^0$ and 6.619 $A^0$, respectively. In addition, the total magnetic moments for these compounds were recorded as 1.9786 $\mu_B$ and 1.99 $\mu_B$, respectively. The conventional $Co_2TiSn$ and inverse $Zr_2RhGa$ compounds had indirect energy gaps of 0.482 eV and 0.573 eV, respectively. From their electronic properties, it can be noted that the conventional full-Heusler $Co_2TiSn$ compound and the inverse full-Heusler $Zr_2RhGa$ compound had stability from a mechanical perspective.

Furthermore, Gupta et al. prepared $Cr_2MnSb$ thin films on a MgO (001) substrate using the DC/RF magnetron sputtering method. The XRD analysis of the deposited films revealed that they crystalized in a cubic phase with full B2 and partial $L2_1$ ordering [25]. Previously, Dubowik et al. deposited 100 nm $Co_2CrAl$ films on glass and NaCl substrates using the flash evaporation technique [26].

Paudel and Zhu [27] showed that the full Heusler alloy $Co_2ScSb$ is stable at the ferromagnetic phase with an optimized lattice constant of 6.19 . They confirmed the structural stability from the calculations of the negative cohesive, formation energy and real phonon frequency. Paudel and Zhu [28] also showed that the half-metallic ferromagnetic properties of a $Fe_2MnP$ alloy have energy band gaps of 0.34 eV and half-metallic gaps of 0.09 eV at an optimized lattice parameter of 5.56 .

In this article, the motivation for investigating the mechanical, electronic and magnetic characteristics of the full-Heusler compounds $Co_2CrAl$ and $Cr_2MnSb$ in both the conventional and inverse form was to study in detail their mechanical and structural stability and preferable magnetic phase, as well as to introduce their elastic properties and behaviors. The article is organized as follows: after the introduction and background review, the

computational methods and model are introduced. This is followed by a presentation of the results, discussion and conclusion.

## 2. Computational Method

In the current study, the calculations were accomplished using the full-potential, linearized augmented plane wave procedure executed in the WIEN2k [29] suite. Generalized gradient approximation (GGA) was used to calculate the structural parameters, i.e., the lattice parameters and bulk modulus. The GGA method depends on the local gradient of the electronic density in addition to the value of the density, giving a more accurate description of variations in the electron–electron interactions. GGA functionals provide a severe underestimation of the energy band gaps, so a modified Becke–Johnson (mBJ-GGA) functional was used to improve the energy band gaps. For the compound $Co_2CrAl$, the muffin-tin radii ($R_{MT}$) of the Co, Cr and Al atoms were taken to be 2.1, 2.05 and 1.95 a.u., respectively, and for the compound $Cr_2MnSb$, the $R_{MT}$ of Cr, Mn and Sb atoms were 2.14, 2.2 and 2.2 a.u., respectively. Moreover, 35 special k-points in the irreducible Brillion Zone (IBZ) with a grid size of $10 \times 10 \times 10$ (equal to 1000 k-points in the Full Brillion Zone (FBZ)) [30] were employed in obtaining self-consistency calculations for the $Co_2CrAl$ and $Cr_2MnSb$ compounds. In addition, the plane waves quantity was limited as $K_{MAX} \times R_{MT} = 8$, and the wave functions' expansions was set by l = 10 inside the muffin-tin spheres. Furthermore, the self-consistent computations were only perceived as well-converged when the computed aggregate crystal energy converged to lower than $10^{-5}$ Ry. In addition, the cubic phase's elastic constants were computed using the second-order derivatives within the WIEN2k-code-contained formalism.

## 3. Results and Discussion

### 3.1. Structural Properties

By fitting the total energy to Murnaghan's equation of state (EOS) [31], the optimized lattice constant (a), bulk modulus (B) and its pressure derivative (B′) were computed as given below:

$$E(\nu) = E_0 + \frac{BV}{B'} \left[ \frac{\left( \frac{V_0}{V} \right)^{B'}}{B' - 1} + 1 \right] - \frac{B'V_0}{B' - 1} \tag{1}$$

where $B$ represents the bulk modulus at the equilibrium volume, $B'$ is the pressure derivative of the bulk modulus at the equilibrium volume and $E_0$ is the minimum energy. The bulk modulus ($B$) and the pressure ($P$) are given by $B = -V \frac{dP}{dV} = V \frac{d^2E}{dV^2}$ and $P = -\frac{dE}{dV}$.

The conventional Heusler $Co_2CrAl$ and $Cr_2MnSb$ compounds had a space group Fm-3m L21 (225) and the inverse Heusler $Co_2CrAl$ and $Cr_2MnSb$ compounds had a space group F-43m Xa (216) [5], while tetragonal crystal lattices were the result of stretching of the cubic lattice along with one of its vectors. This resulted in the cube taking the shape of a rectangular prism whose base was a square (a by a), and the height (c) was different from the base edge a. Therefore, the tetragonal Heusler $Co_2CrAl$ and $Cr_2MnSb$ compounds had space groups of I4/mmm (139) and I-4m2 (119). The full-Heusler structural properties of the $Co_2CrAl$ and $Cr_2MnSb$ compounds were calculated. Figure 1 presents the crystal structures of the full-Heusler $Co_2CrAl$ and $Cr_2MnSb$ compounds. The aggregate energy as a function of the volume for the Heusler $Co_2CrAl$ and $Cr_2MnSb$ compounds are presented in Figures 2 and 3. Moreover, the state (EOS) was used to compute the optimized structural parameters, presented in Tables 1 and 2.

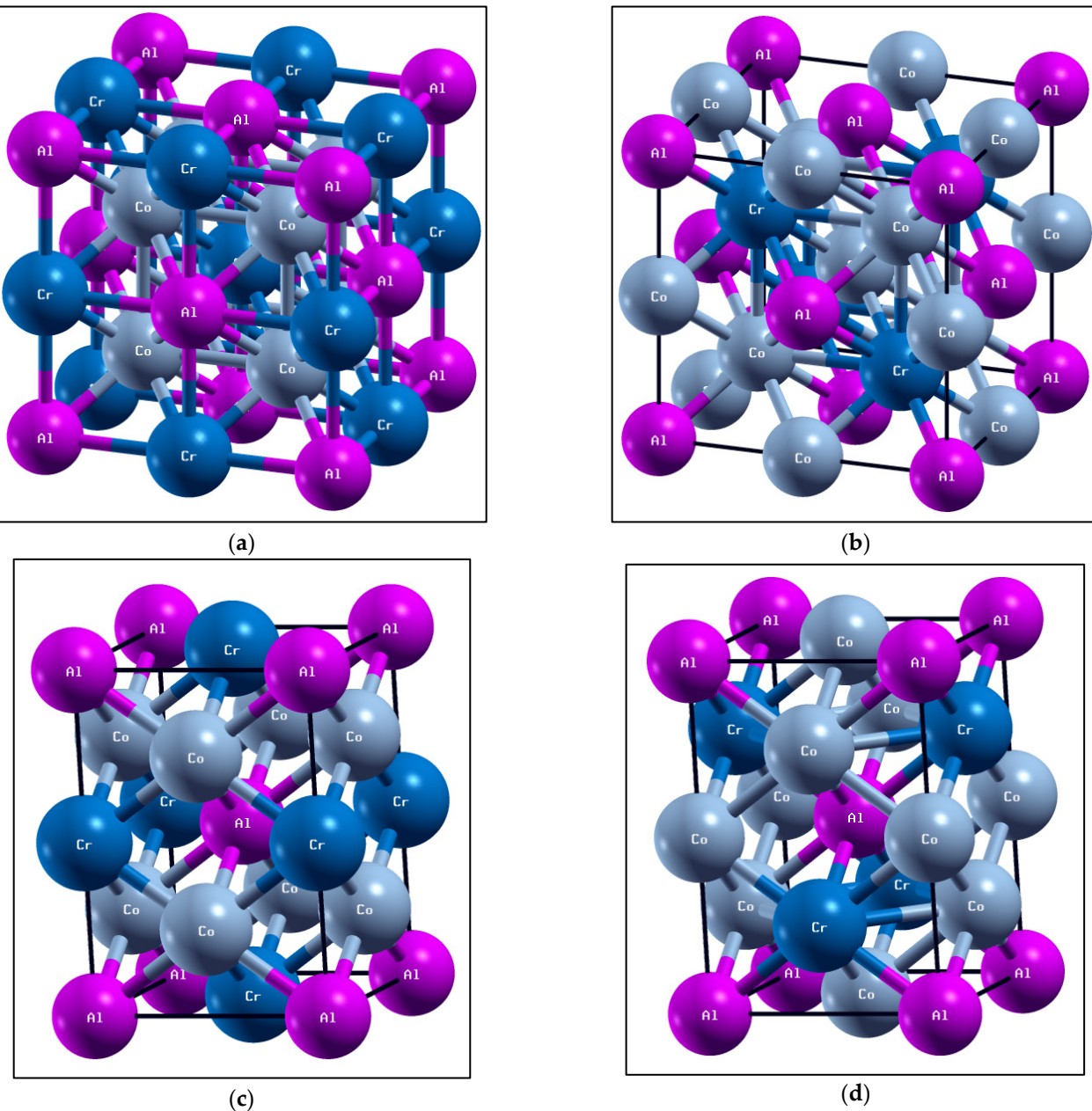

**Figure 1.** Different crystal structures of Heusler Co$_2$CrAl. (**a**) Conventional Heusler structure Co$_2$CrAl Fm-3m L21 (225), (**b**) inverse Heusler Co$_2$CrAl structure F-43m X (216), (**c**) tetragonal structure I4/mmm (139) and (**d**) tetragonal structure I-4m2 (119).

Tables 1 and 2 show our computed lattice parameters compared with other theoretical and experimental lattice parameters of conventional and inverse Heusler Co$_2$CrAl and Cr$_2$MnSb compounds. The calculated lattice parameters for the conventional Heusler Co$_2$CrAl compound deviated from the measured one within 0.38% [19]. The calculated lattice parameters for the conventional and inverse Heusler Cr$_2$MnSb compounds perfectly agreed with the theoretical outcomes [21]. As far as we know, comparable experimental results for conventional and inverse Heusler Cr$_2$MnSb compounds are not available. These results ensured the reliability of the present first-principle computations.

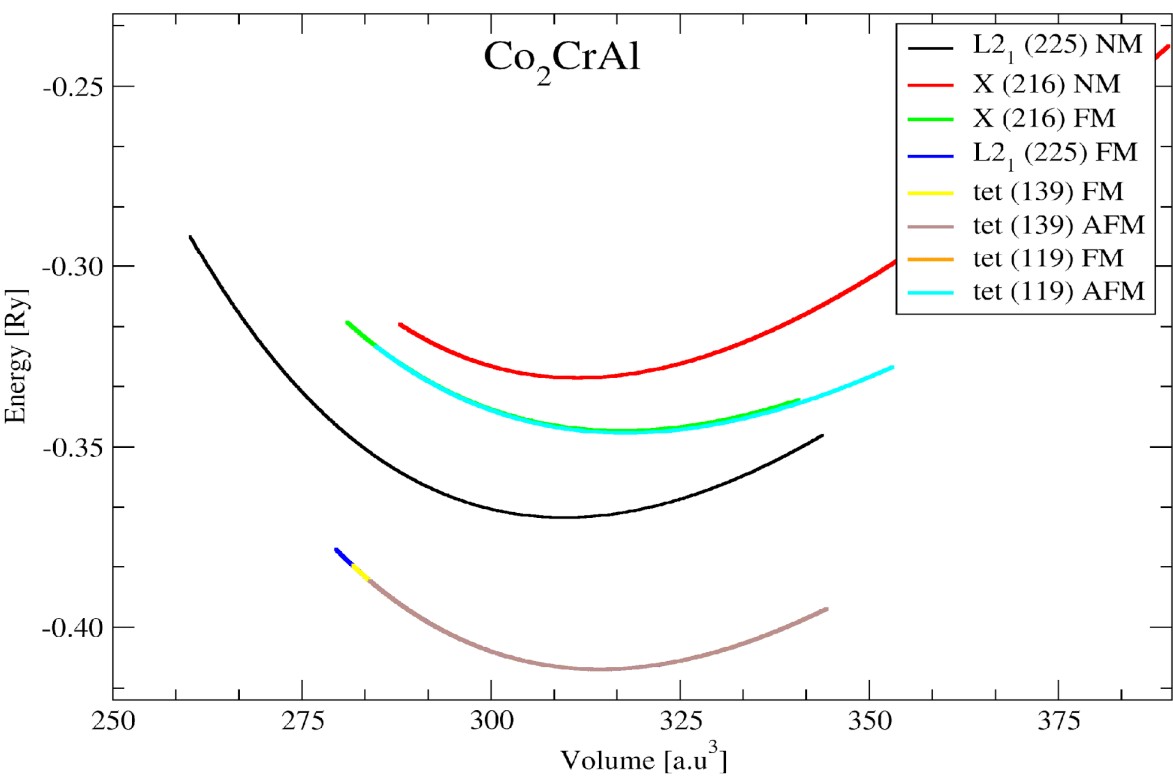

**Figure 2.** The total energy (Ry) versus volume (a.u.$^3$) for different crystal structures of Heusler Co$_2$CrAl.

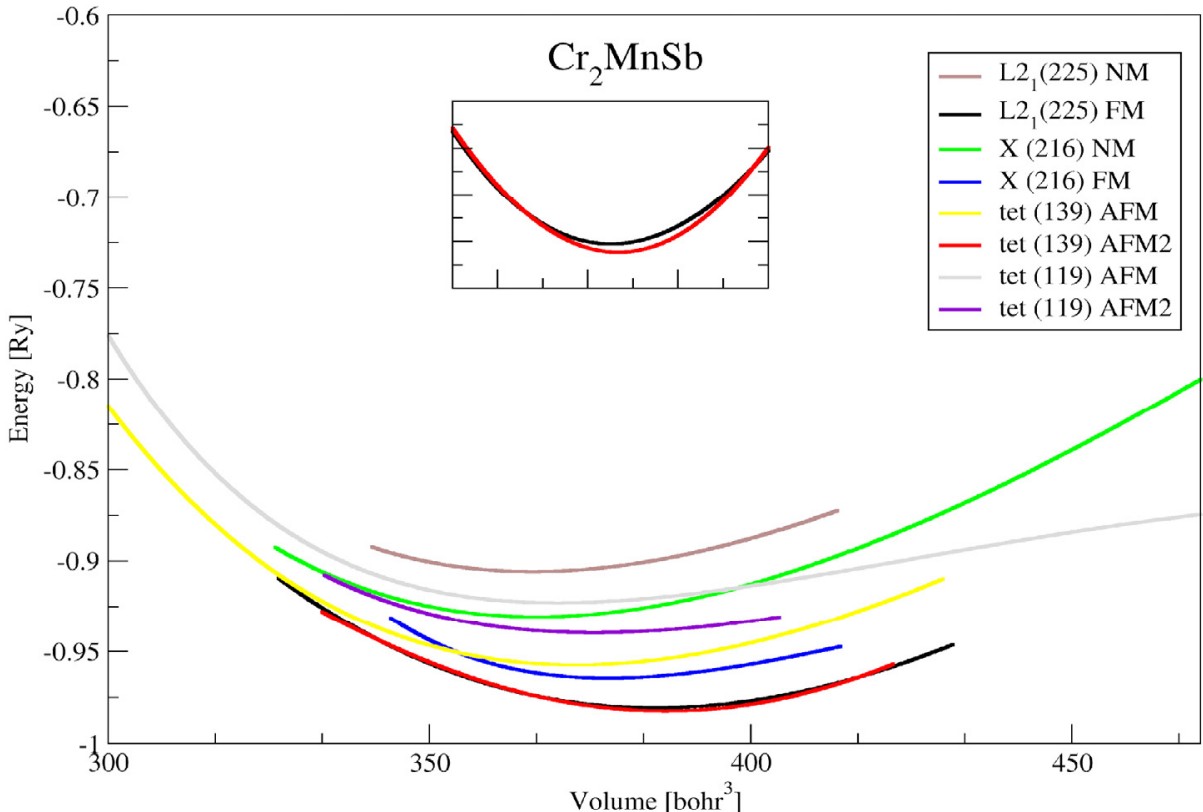

**Figure 3.** The total energy (Ry) versus volume (a.u.$^3$) for different crystal structures of Heusler Cr$_2$MnSb.

**Table 1.** Calculated lattice parameter (a), bulk modulus (B), and total energy ($E_{tot}$) for Heusler $Co_2CrAl$ compound.

| Structure | Space Group | Magnetic Phase | Reference | Lattice Parameter a (Å) | B (GPa) | $E_{total}$ (Ry)/f.u | c/a |
|---|---|---|---|---|---|---|---|
| $Co_2CrAl$ | Conventional Fm-3m (225) | NM | Present | 5.6830 | 213.468 | −8161.3694 | 1 |
| | Conventional Fm-3m (225) | FM | Present | 5.7082 | 206.811 [19] | −8161.4115 [19] | 1 |
| | | | Experimental | 5.74 [19] 5.70 [26] | | | |
| | | | Theoretical | 5.73 [20] | | | |
| | Inverse F-43m (216) | NM | Present | 5.6936 | 212.056 | −8161.3309 | 1 |
| | Inverse F-43m (216) | FM | Present | 5.7398 | 169.0397 | −8161.3454 | 1 |
| | I4/mmm (139) | FM | Present | 4.011 | 202.24 | −8161.41148 | 1.4175 |
| | I4/mmm (139) | AFM | Present | 4.0239 | 202.133 | −8161.4095 | 1.4175 |
| | I-4m2 (119) | FM | Present | 3.9382 | 166.847 | −8161.34608 | 1.5253 |
| | I-4m2 (119) | AFM | Present | 3.9441 | 163.781 | −8161.3458 | 1.5253 |

**Table 2.** Calculated lattice parameter (a), bulk modulus (B), and total energy ($E_{tot}$) for Heusler $Cr_2MnSb$ compound.

| Structure | Space Group | Magnetic Phase | Reference | Lattice Parameter a (Å) | B (GPa) | $E_{total}$ (Ry)/f.u | c/a |
|---|---|---|---|---|---|---|---|
| $Cr_2MnSb$ | Conventional Fm-3m (225) | NM | Present | 6.1116 | 201.77 | −19,487.9803 | 1 |
| | Conventional Fm-3m (225) | FM | Present | 6.1381 | 248.45 | −19,487.98197 | 1 |
| | | | Theoretical Experimental | 6.0 [21] 5.95 [25] | | | |
| | Inverse F-43m (216) | NM | Present | 6.0724 | 220.322 | −19,487.9644 | 1 |
| | Inverse F-43m (216) | FM | Present | 6.0571 | 149.5381 | −19,487.9683 | 1 |
| | | | Theoretical Result | 5.9 [21] | | | |
| | I4/mmm (139) | FM | Present | 4.3337 | 296.852 | −19,487.982 | 1.4158 |
| | I4/mmm (139) | AFM | Present | 4.2637 | 386.728 | −19,487.9823 | 1.4158 |
| | I-4m2 (119) | FM | Present | 4.0617 | 142.459 | −19,487.9394 | 1.6513 |
| | I-4m2 (119) | AFM | Present | 4.0459 | 153.8705 | −19,487.9394 | 1.6513 |

According to the results obtained in this study, our volume optimization results showed that AFM tetragonal distortion (No. 139) was more preferred than FM cubic $L2_1$ for the $Cr_2MnSb$ compound with a slightly small energy difference $\Delta E_{tet-cubic} = 0.0047R\frac{Ry}{f.u}$ (see Equation (2)). On the other hand, FM cubic $L2_1$ was more preferred than tetragonal distortion for the $Co_2CrAl$ case, with an energy difference $\Delta E_{tet-cubic} = 0.002\frac{Ry}{f.u}$. To make the AFM tetragonal phase stable, the energy difference with a cubic structure should be greater than 0.1 eV/f.u. As reported previously, $Cr_2MnSb$ crystallizes in a cubic $L2_1$ structure with a fully compensated ferrimagnetic configuration, where the magnetic moment of Cr and Mn are dominated by antiparallel exchange [21].

$$\Delta E_{tet-cubic} = E_{tet} - E_{cubic} \tag{2}$$

*3.2. Magnetic Properties*

This part involved the calculation of the inverse, conventional and tetragonal I4/mmm (139) Heusler $Co_2CrAl$ and $Cr_2MnSb$ compounds' partial and total magnetic moments. The results obtained were compared with other theoretical values as shown in Tables 3 and 4.

**Table 3.** Total magnetic moment for inverse, conventional and tetragonal I4/mmm (139) Heusler $Co_2CrAl$ compound.

| Compounds | | Magnetic Moment in $\mu_B$ | | | | | |
|---|---|---|---|---|---|---|---|
| | | Co | Co | Cr | Al | Interstitial | Total Magnetic Moment ($M^{tot}$) in $\mu_B$ |
| Inverse $Co_2CrAl$ | Present | 0.96069 | 1.36311 | −1.26906 | −0.00841 | −0.21497 | 0.83116 |
| Conventional $Co_2CrAl$ | Present | 1.01815 | 1.01815 | 1.32570 | −0.06082 | −0.30118 | 3 |
| | Theoretical Result | 0.650 [19] | 0.650 [19] | 1.745 [19] | −0.045 [19] | – | 3 [19] |
| | Theoretical Result | – | – | – | – | – | 2.96 [22] |
| Tetragonal I4/mmm (139) $Co_2CrAl$ | Present | −0.04084 | −0.04084 | 0.81717 | 1.45741 | −0.05097 | 2.99994 |

**Table 4.** Total magnetic moment for inverse, conventional and tetragonal I4/mmm (139) Heusler $Cr_2MnSb$ compound.

| Compounds | | Magnetic Moment in $\mu_B$ | | | | | |
|---|---|---|---|---|---|---|---|
| | | Cr | Cr | Mn | Sb | Interstitial | Total Magnetic Moment ($M^{tot}$) in $\mu_B$ |
| Inverse $Cr_2MnSb$ | Present | −1.72053 | 2.68899 | −1.05810 | 0.04777 | 0.04187 | 0 |
| | Theoretical Result | 1.96 [21] | −3.18 [21] | 1.29 [21] | - | - | 0 [21] |
| Conventional $Cr_2MnSb$ | Present | −1.51854 | −1.51854 | 3.21064 | 0.06167 | −0.23513 | 0.00011 |
| | Theoretical Result | 1.77 [21] | 1.77 [21] | −3.44 [21] | - | - | 0.01 [21] |
| Tetragonal I4/mmm (139) $Cr_2MnSb$ | Present | 0.05294 | 0.05294 | −1.46141 | 3.08253 | −0.20953 | 0.00312 |

We found that the conventional and tetragonal Heusler $Co_2CrAl$ compounds were ferromagnetic compounds. Furthermore, the total magnetic moment for the inverse $Co_2CrAl$ compound was $M^{tot} = 0.83116\ \mu_B$, while it was $M^{tot} = 3\ \mu_B$ for the conventional $Co_2CrAl$ compound. The physics interpretation behind this huge difference between the total spin magnetic moment of the conventional and inverse $Co_2CrAl$ was due to antiparallel exchange interactions between the Cr atom and Co atom in the case of inverse $Co_2CrAl$, whereas it was a direct interaction in the case of the conventional phase. Therefore, it can be noted from the results produced here that the conventional $Co_2CrAl$ compound's computed total magnetic moment perfectly matched with the prior theoretical results [19,22], as Table 3 shows. Theoretically, a compound with a total magnetic moment $M^{tot}$ with an integer value means it is a half-metallic material.

Table 4 shows the results for the inverse, conventional and tetragonal I4/mmm (139) Heusler $Cr_2MnSb$ compounds. Table 4 shows that the inverse Heusler $Cr_2MnSb$ had a negative spin moment for one Cr atom and one Mn atom and a positive spin moment for the second Cr atom. The conventional Heusler $Cr_2MnSb$ had a negative spin moment for its Cr atoms and a positive spin moment for its Mn atom. The Sb atom's spin moment was extremely tiny in both structural types of $Cr_2MnSb$. The electronic configurations in the Mn and Cr atoms were similar, and they had one electron difference. Consequently, their exchange maintained the compound's ferromagnetic character, which led to small variations in the spin moments per site.

We found that the conventional Heusler $Cr_2MnSb$ had a small total magnetic moment (non-zero total magnetization) due to the decrease in the atomic disorder in the Mn–Sb sublattice. This implied that the conventional Heusler $Cr_2MnSb$ compound had a ferrimagnetic order.

On the other hand, we found that the inverse Heusler $Cr_2MnSb$ had a zero total magnetic moment, which meant that this compound had an antiferromagnetic magnetic order. The tetragonal $Cr_2MnSb$ had a small total magnetic moment, which meant that this compound was ferrimagnetic.

### 3.3. Electronic Properties

In this section, the partial and total density of states and the band structure for the inverse and conventional Heusler ($Co_2CrAl$, $Cr_2MnSb$) compounds were investigated. An analysis of the density of states and band structure showed that the conventional $Co_2CrAl$, conventional $Cr_2MnSb$ and inverse $Cr_2MnSb$ Heusler compounds exhibited a half-metallic conduct in a ferromagnetic state. This implied that the spin-up electrons in the materials had a metallic behavior; when they behaved as semiconducting with a spin-down direction, the materials had a semiconducting behavior. On the other hand, the inverse Heusler $Co_2CrAl$ compound had a metallic behavior. The tetragonal I4/mmm (139) Heusler ($Co_2CrAl$, $Cr_2MnSb$) compounds had a half-metallic behavior in the antiferromagnetic state.

Figure 4a,b shows the metallic behavior of the spin up and spin down band structures within the PBE-GGA method for the inverse Heusler $Co_2CrAl$ compound with zero energy gap. Figure 5a,b also shows the metallic behavior of the spin-up and spin-down band structures within the mBJ-GGA method for the inverse Heusler $Co_2CrAl$ compound with zero energy gap.

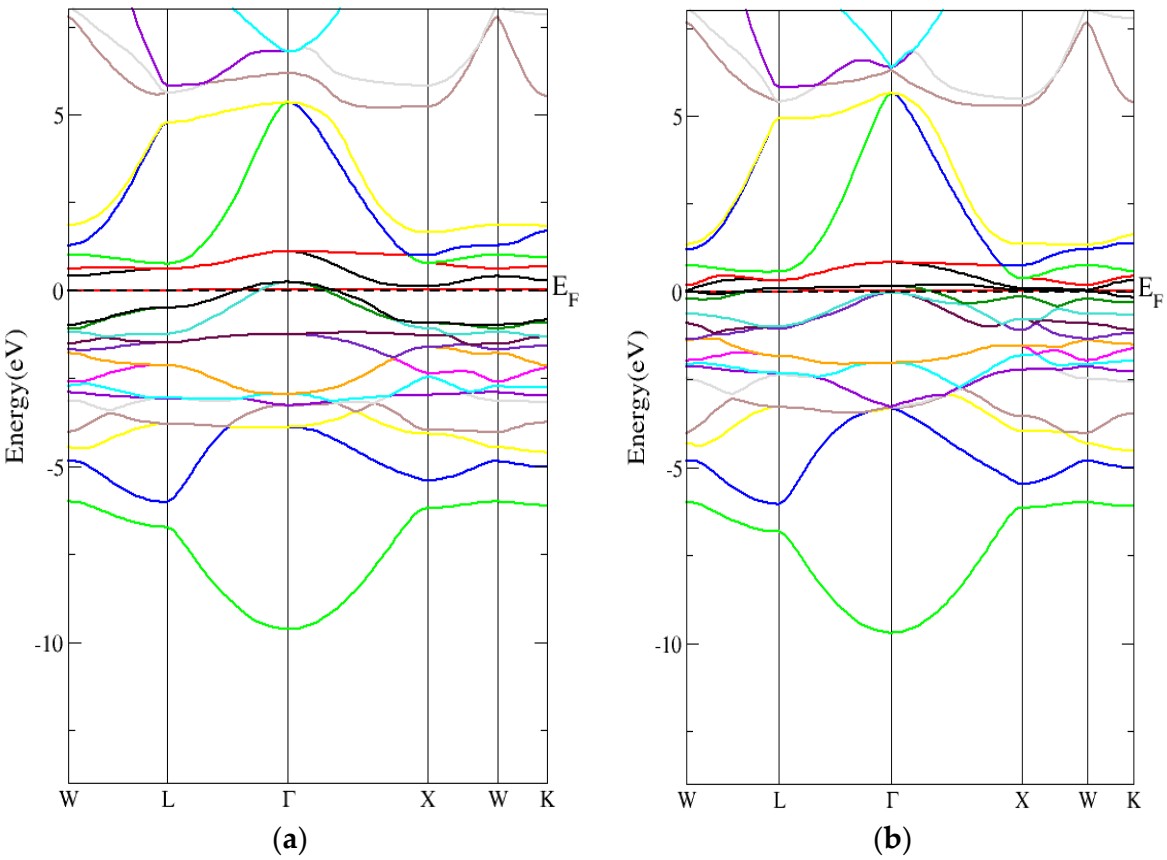

**Figure 4.** The band structure for the inverse Heusler $Co_2CrAl$ compound by employing the PBE-GGA technique for (**a**) spin-up inverse Heusler $Co_2CrAl$ compound and (**b**) spin-down inverse Heusler $Co_2CrAl$ compound.

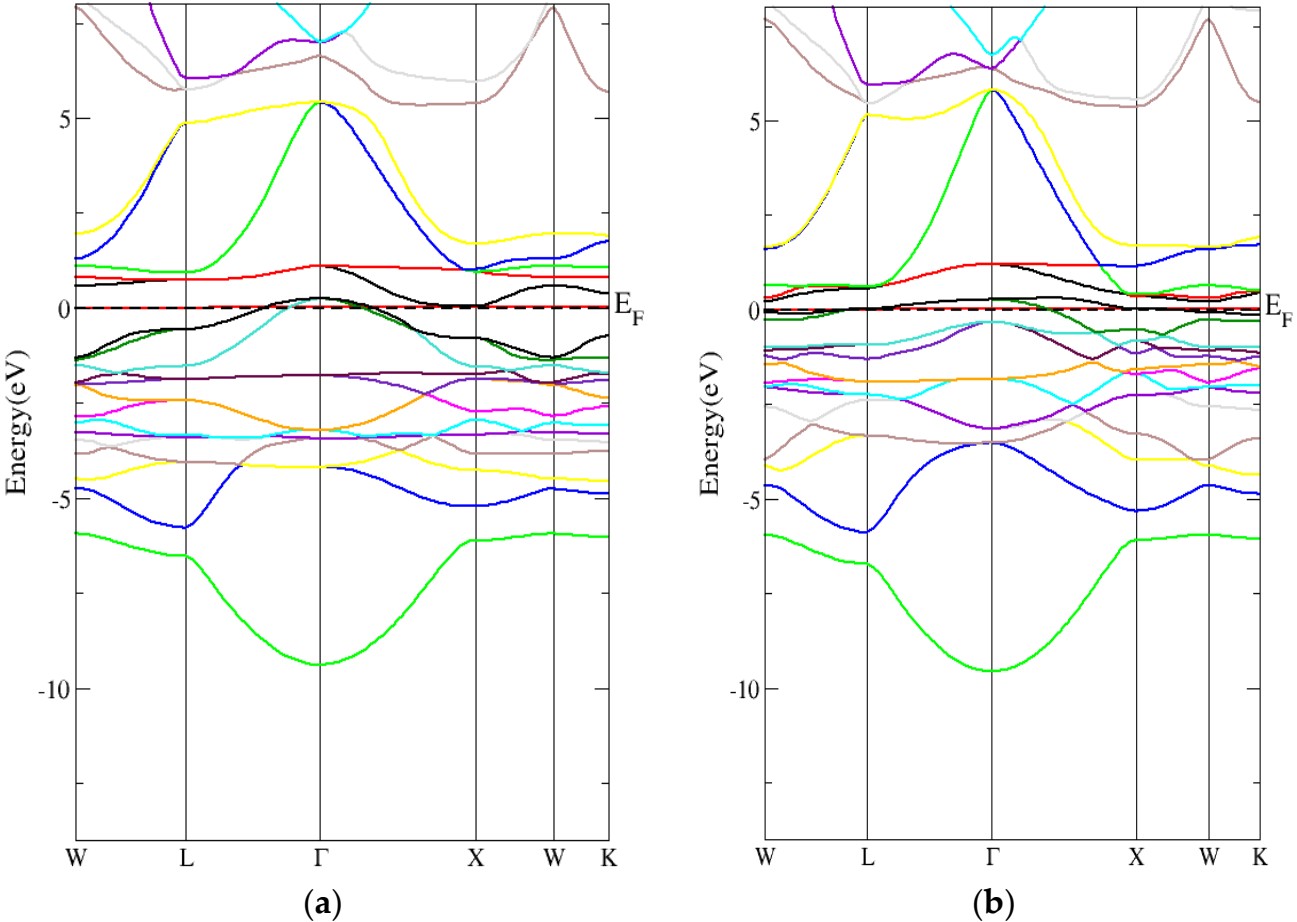

**Figure 5.** The band structure for the inverse Heusler Co$_2$CrAl compound by employing the mBJ-GGA technique for (**a**) spin-up inverse Heusler Co$_2$CrAl compound (**b**) spin-down inverse Heusler Co$_2$CrAl compound.

From Figure 6a,b, the conventional Heusler Co$_2$CrAl compound's band structure had an indirect energy band gap (spin down) using the PBE-GGA technique. In addition, Figure 7a,b shows that the conventional Heusler Co$_2$CrAl compound's band structure had an indirect energy band gap (spin-down) when using the mBJ-GGA technique. As indicated in Table 5, the indirect energy gaps within PBE-GGA and mBJ-GGA were 0.6 eV and 0.9 eV, respectively.

**Table 5.** The energy band gaps for conventional and inverse Co$_2$CrAl compound using PBE-GGA and mBJ methods.

| Compounds | Band Gap Type | High Symmetry Lines | $E_g$-PBE-GGA (eV) | $E_g$-mBJ-GGA (eV) |
|---|---|---|---|---|
| Conventional $-$Co$_2$CrAl | Indirect | $\Gamma - X$ | 0.6 | 0.9 |
| Inverse $-$ Co$_2$CrAl | Metallic | - | - | - |

From Figure 8a,b, the band structure (spin-down) of the inverse Heusler Cr$_2$MnSb compound had a direct energy band gap using the PBE-GGA technique. In addition, Figure 9a,b shows that the inverse Heusler Cr$_2$MnSb compound's band structure (spin-down) had a direct energy band gap when the mBJ-GGA technique was used. As can be seen in Table 6, the direct energy gaps within the PBE-GGA and mBJ-GGA methods were 0.8 eV and 0.9 eV, respectively.

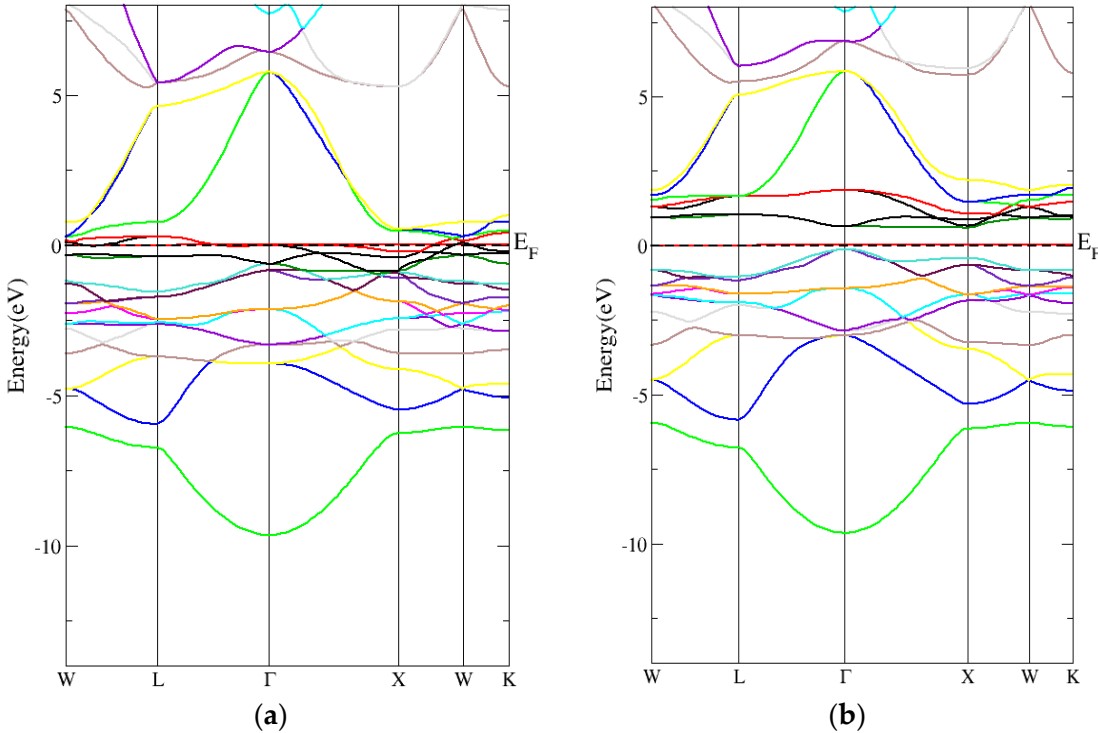

**Figure 6.** The band structure for the conventional Heusler Co$_2$CrAl compound by employing the PBE-GGA technique for (**a**) spin-up conventional Heusler Co$_2$CrAl compound and (**b**) spin-down conventional Heusler Co$_2$CrAl compound.

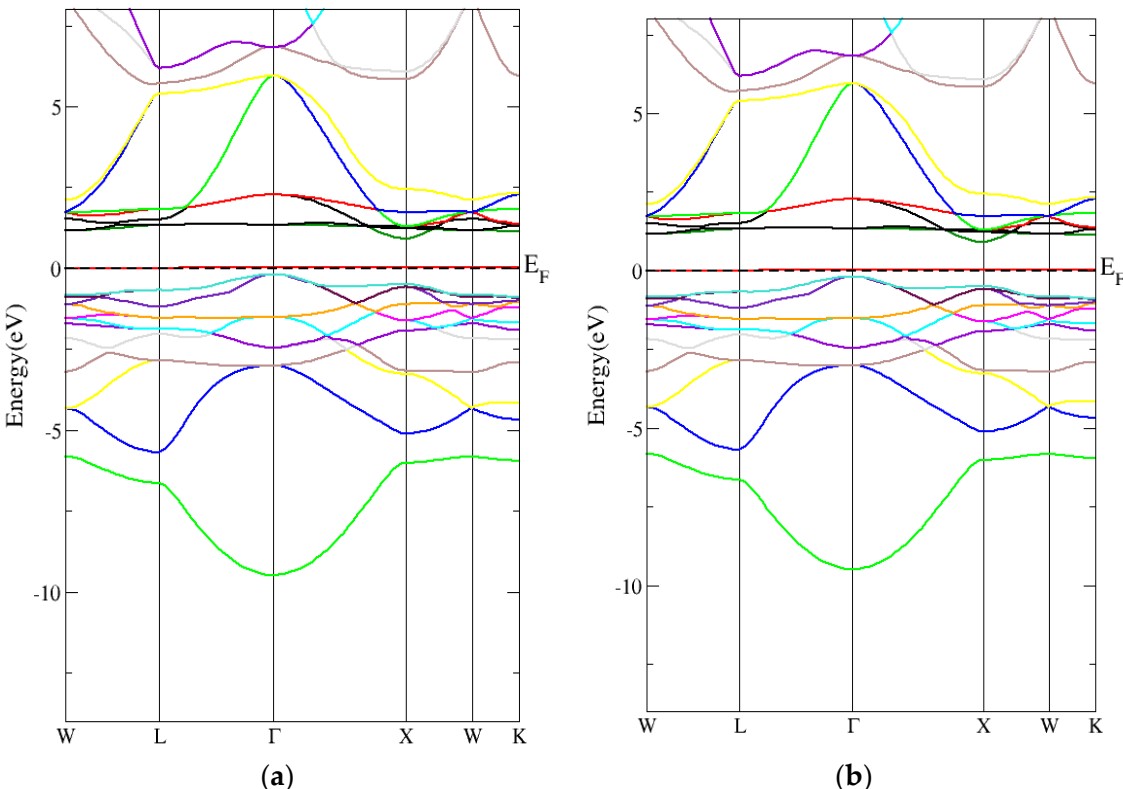

**Figure 7.** The band structure for the conventional Heusler Co$_2$CrAl compound by employing the mBJ-GGA technique for (**a**) spin-up conventional Heusler Co$_2$CrAl compound and (**b**) spin-down conventional Heusler Co$_2$CrAl compound.

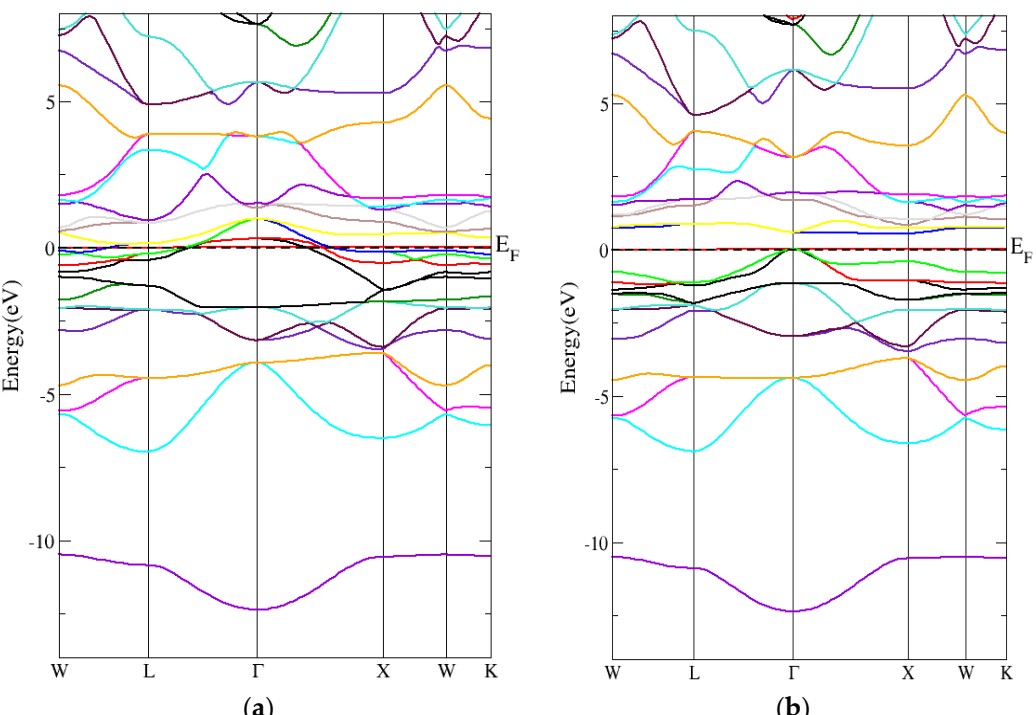

**Figure 8.** The band structure for the inverse Heusler $Cr_2MnSb$ compound by employing the PBE-GGA technique for (**a**) spin-up inverse Heusler $Cr_2MnSb$ compound and (**b**) spin-down inverse Heusler $Cr_2MnSb$ compound.

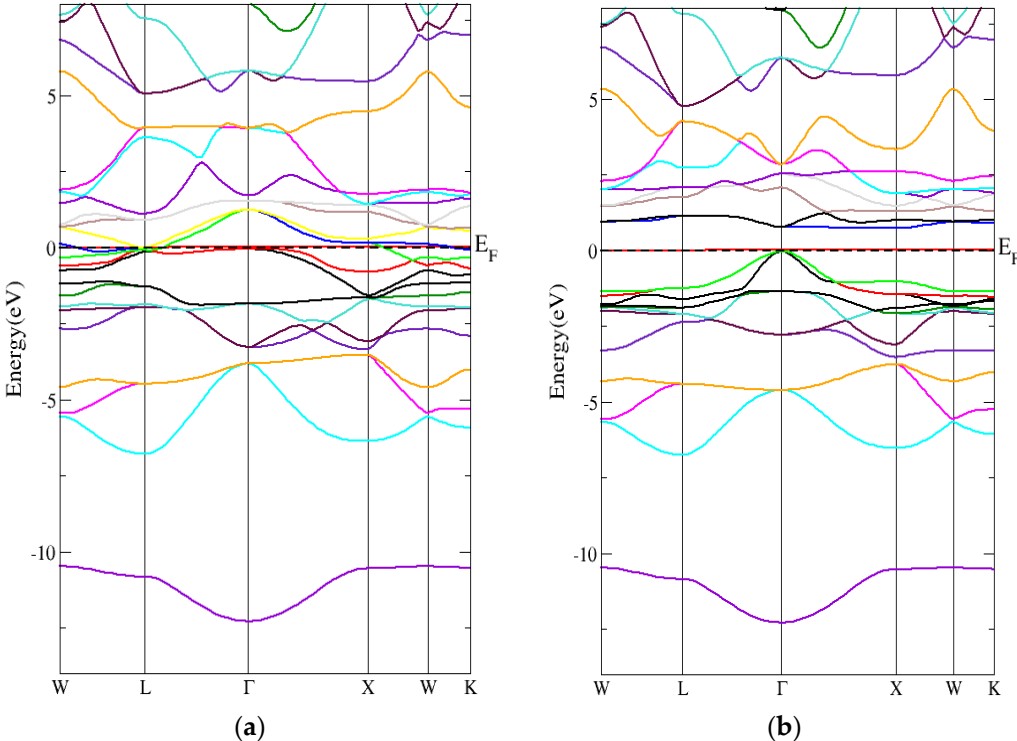

**Figure 9.** The band structure for the inverse Heusler $Cr_2MnSb$ compound by employing the mBJ-GGA technique for (**a**) spin-up inverse Heusler $Cr_2MnSb$ compound and (**b**) spin-down inverse Heusler $Cr_2MnSb$ compound.

**Table 6.** The energy band gaps for conventional and inverse $Cr_2MnSb$ compound using PBE-GGA and mBJ methods.

| Compounds | Band Gap Type | High Symmetry Lines | $E_g$-PBE-GGA (eV) | $E_g$-mBJ-GGA (eV) |
|---|---|---|---|---|
| Conventional-$Cr_2MnSb$ | Direct | $\Gamma$ | 0.9 | 1 |
| Inverse-$Cr_2MnSb$ | Direct | $\Gamma$ | 0.8 | 0.9 |

Figure 10a,b shows that the band structure of the conventional Heusler $Cr_2MnSb$ compound had a direct energy band gap (spin-down) using the PBE-GGA technique. In addition, Figure 11a,b shows that the Heusler $Cr_2MnSb$ compound's band structure had a direct energy band gap (spin-down) using the mBJ-GGA technique. As indicated in Table 6, the indirect energy gaps within PBE-GGA and mBJ-GGA methods were 0.9 eV and 1 eV, respectively.

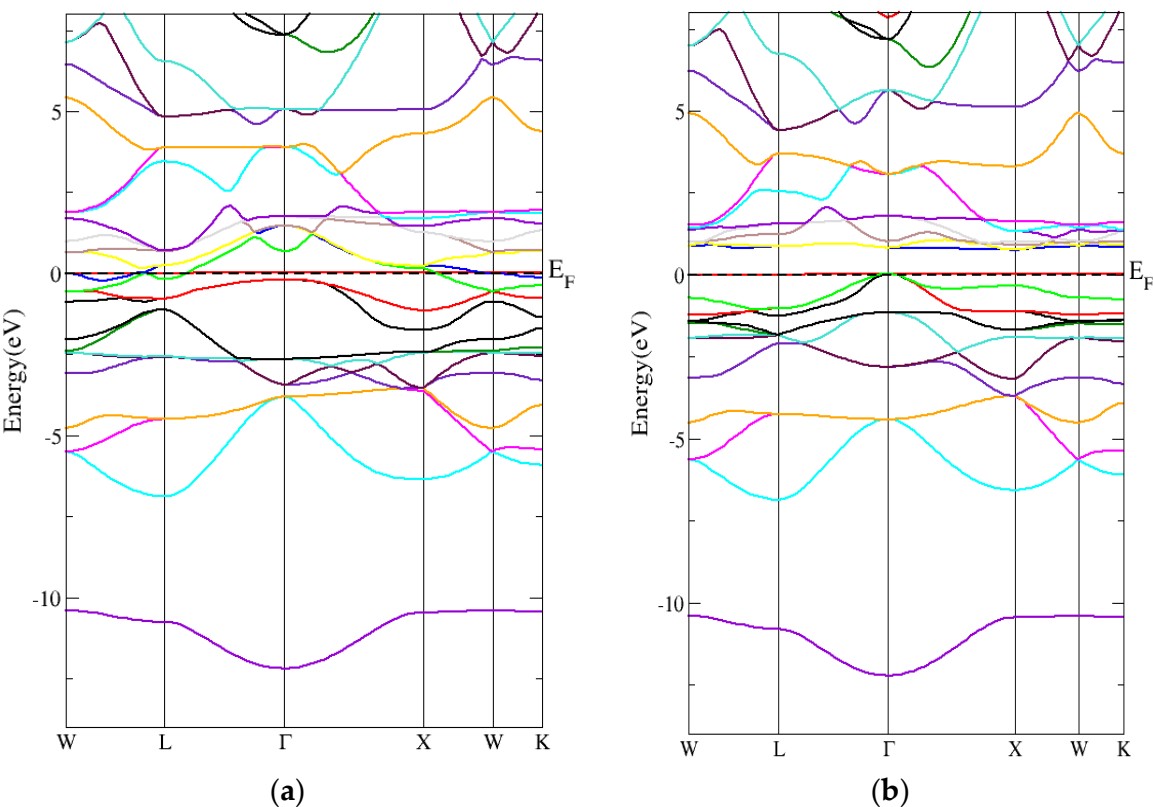

**(a)**        **(b)**

**Figure 10.** The band structure for the conventional Heusler $Cr_2MnSb$ compound by employing the PBE-GGA technique for (**a**) spin-up conventional Heusler $Cr_2MnSb$ compound and (**b**) spin-down conventional Heusler $Cr_2MnSb$ compound.

Figure 12a,b presents the Heusler $Co_2CrAl$ compound band structure for tetragonal I4/mmm (139) in the AFM state. Figure 12a shows that the tetragonal I4/mmm (139) Heusler $Co_2CrAl$ compound's spin-up band structure had a metallic character, while Figure 12b indicates that the tetragonal I4/mmm (139) Heusler $Co_2CrAl$ compound's spin-down band structure had a direct energy band gap. The direct energy gap was found to be 0.8 eV, as shown in Table 7. Figure 13a,b shows the tetragonal I4/mmm (139) Heusler $Cr_2MnSb$ compound's band structure in the AFM state. Figure 13a indicates that the tetragonal I4/mmm (139) Heusler $Cr_2MnSb$ compound's spin-up band structure had a metallic nature, while Figure 13b illustrates that the tetragonal I4/mmm (139) Heusler $Cr_2MnSb$ compound's spin-down band structure had a direct energy band gap. The direct energy gap was found to be 0.9 eV, as shown in Table 7.

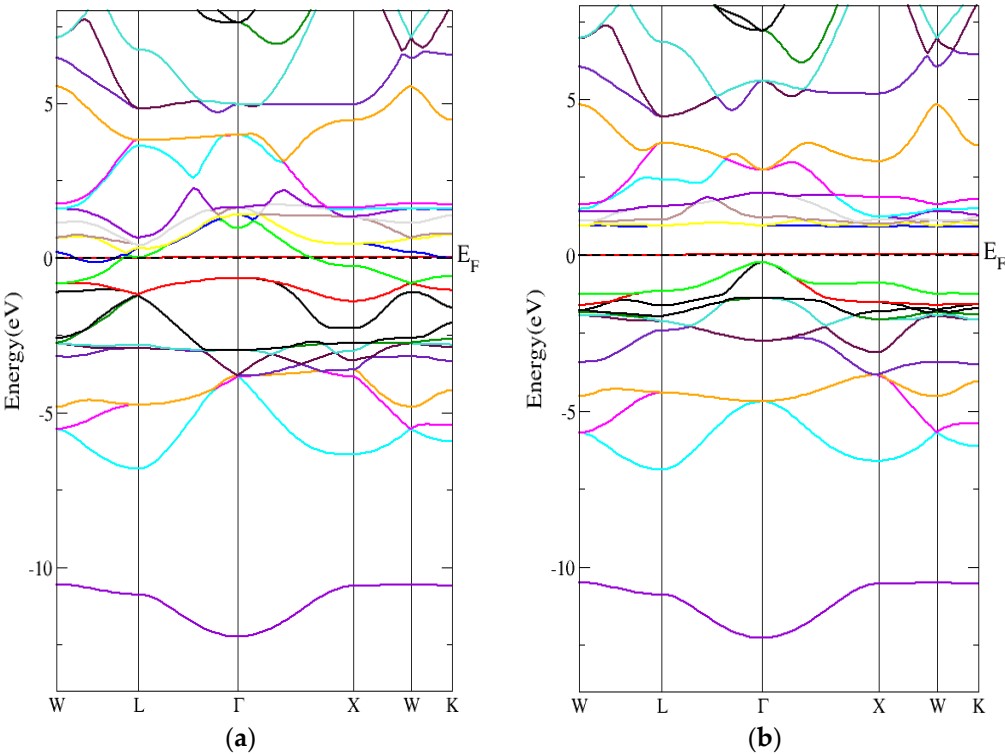

**Figure 11.** The band structure for the conventional Heusler Cr$_2$MnSb compound by employing the mBJ-GGA technique for (**a**) spin-up conventional Heusler Cr$_2$MnSb compound and (**b**) spin-down conventional Heusler Cr$_2$MnSb compound.

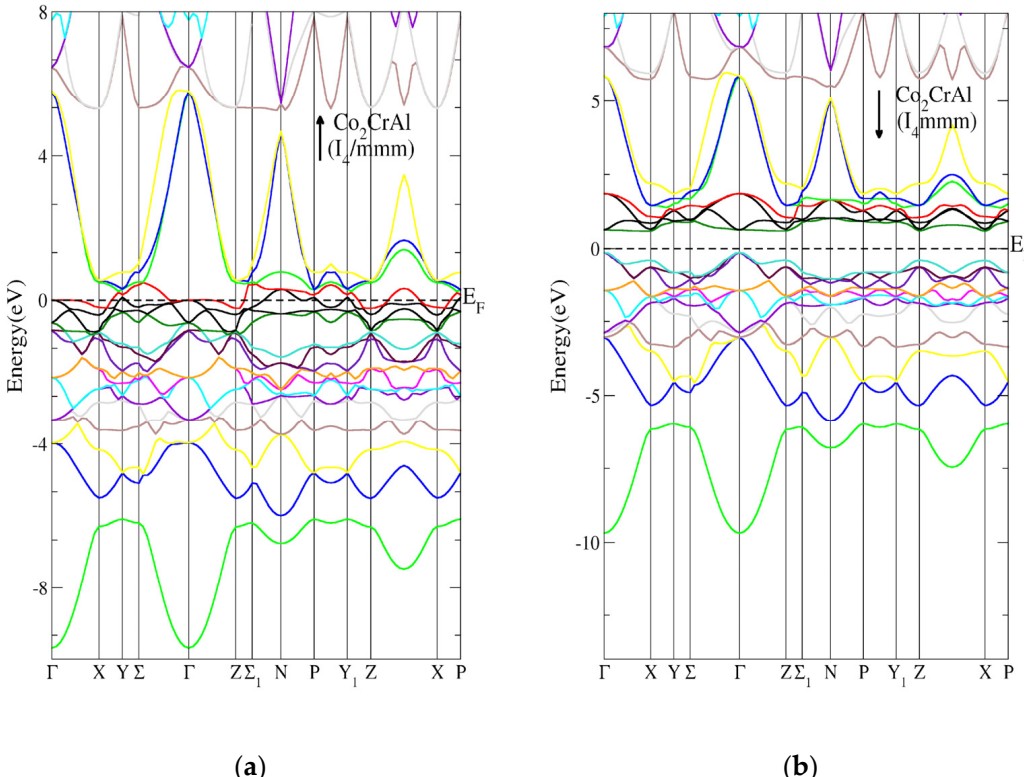

**Figure 12.** The band structure for the tetragonal I4/mmm (139) Heusler Co$_2$CrAl compound in AFM state. (**a**) Spin-up tetragonal I4/mmm Heusler (139) Co$_2$CrAl compound and (**b**) spin-down tetragonal I4/mmm Heusler (139) Co$_2$CrAl compound.

**Table 7.** The energy band gaps for the tetragonal I4/mmm (139) Heusler Co$_2$CrAl and Cr$_2$MnSb compounds in AFM state.

| Compounds | Band Gap Type | High Symmetry Lines | $E_g$(eV) |
|---|---|---|---|
| Co$_2$CrAl | Direct | Γ | 0.8 |
| Cr$_2$MnSb | Direct | Γ | 0.9 |

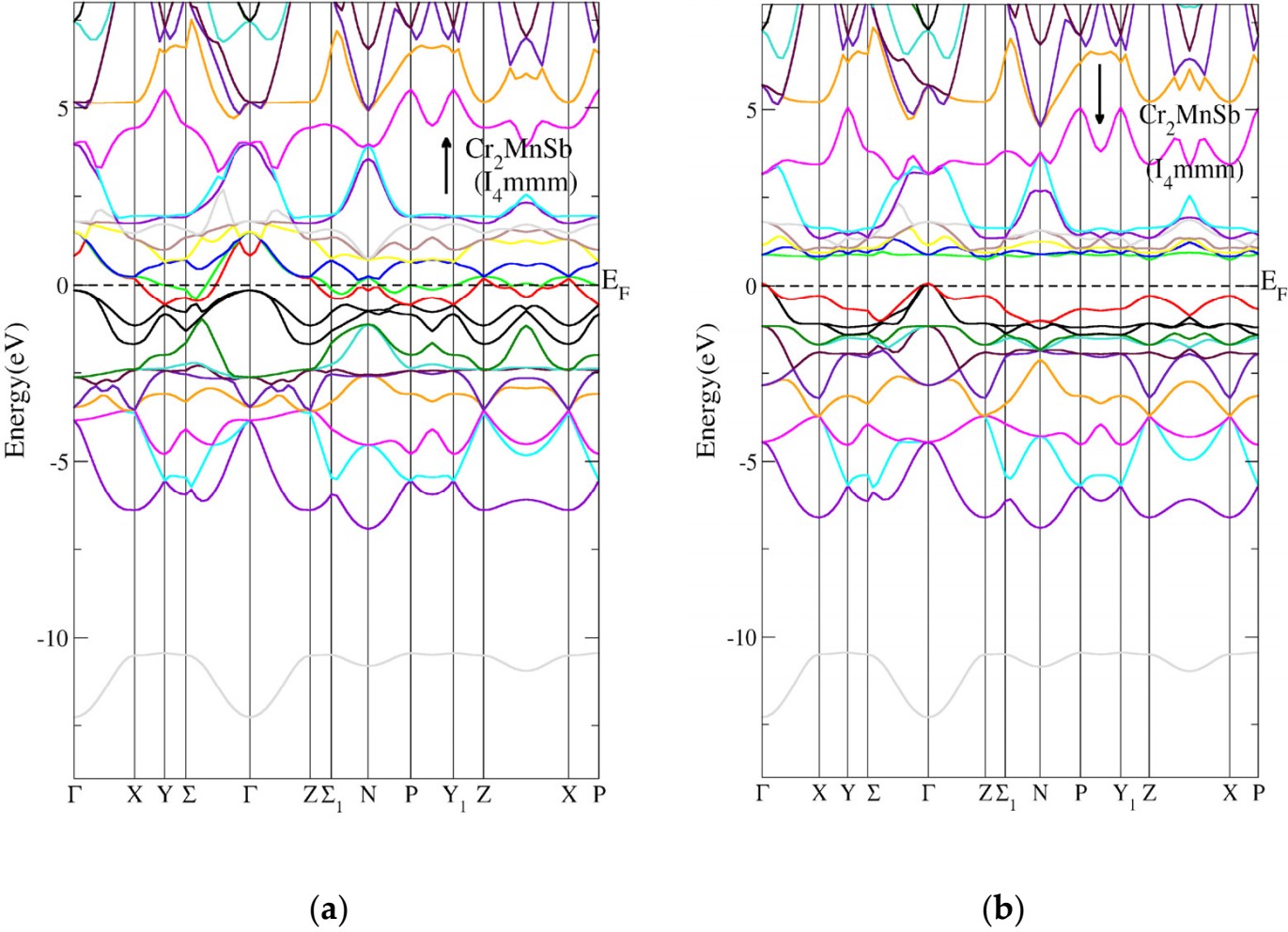

(**a**)  (**b**)

**Figure 13.** The band structure for the tetragonal I4/mmm (139) Heusler Cr$_2$MnSb compound in AFM state. (**a**) Spin-up tetragonal I4/mmm Heusler (139) Cr$_2$MnSb compound and (**b**) spin-down tetragonal I4/mmm Heusler (139) Cr$_2$MnSb compound.

For the conventional and inverse Heusler Co$_2$CrAl and Cr$_2$MnSb compounds, the partial and total density of states for the spin-down, spin-up and inverse Heusler Co$_2$CrAl and Cr$_2$MnSb compounds are presented in Figures 14–17. The density of states in Figures 14–17 also show half-metallic behaviors for the conventional Heusler Co$_2$CrAl and the inverse and conventional Heusler Cr$_2$MnSb compounds with a minor energy band gap in the spin-down segment. This implied that the behavior of these compounds was half-metallic.

In the conventional Co$_2$CrAl′s (Figure 15) spin-down segment, the valence band resulted from the d-state of Co, the d-state of Cr and the tiny effect of the Al in the s-state. The d-state of Co, the d-state of Cr and the tiny effect of the Al in the s-state were accredited to the conduction band. In the spin-down channel of the conventional Co$_2$CrAl, the valence band was attributed to the d-state of Co, the d-state of Cr and the minor effect of Al in the s-state. On the other hand, the conduction band resulted from the d-state of Co, the d-state of Cr and the minor effect of the Al in the s-state.

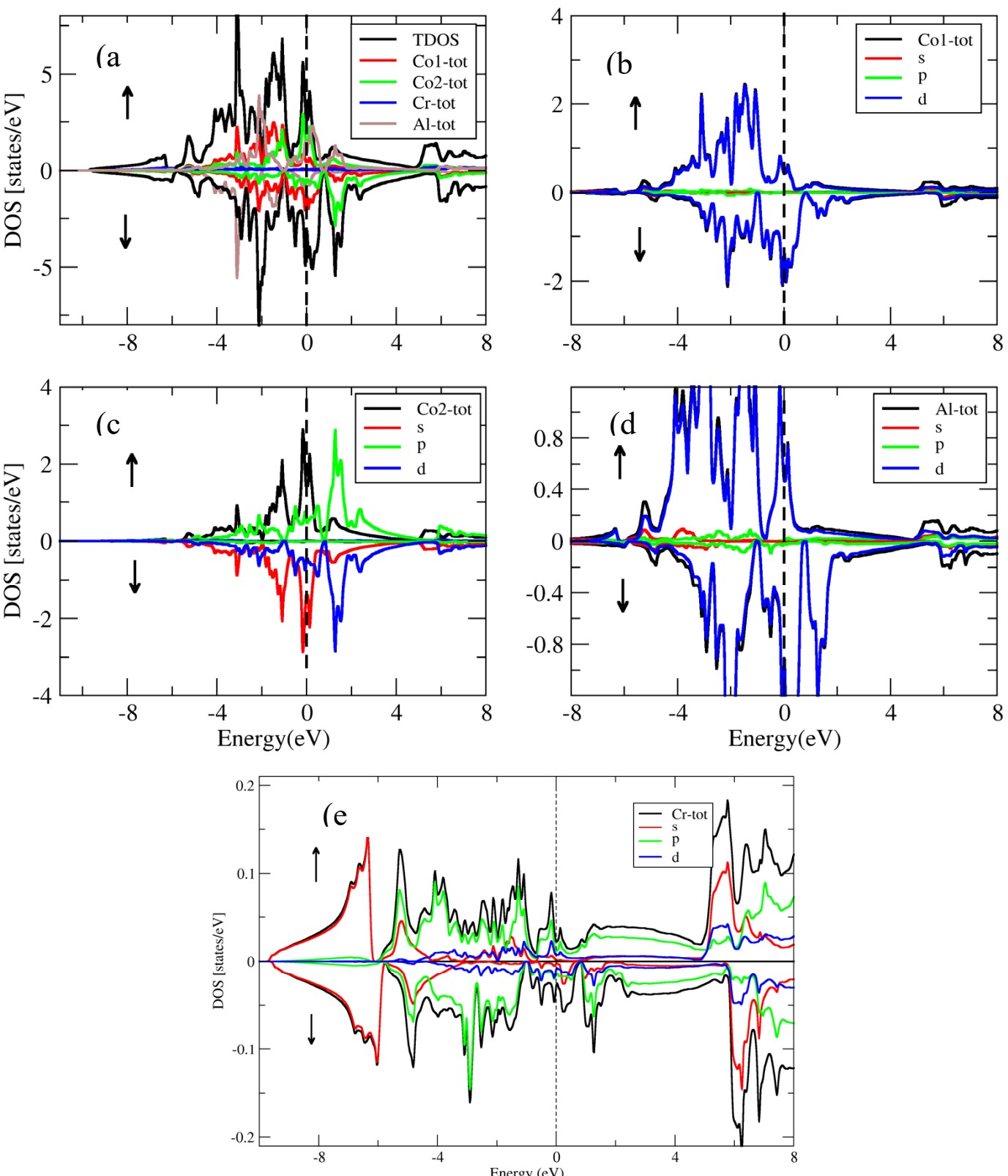

**Figure 14.** (**a**) Total density of states for the inverse $Co_2CrAl$ compound and partial density of states for (**b**) Co1 atom, (**c**) Co2 atom, (**d**) Al atom and (**e**) Cr atom.

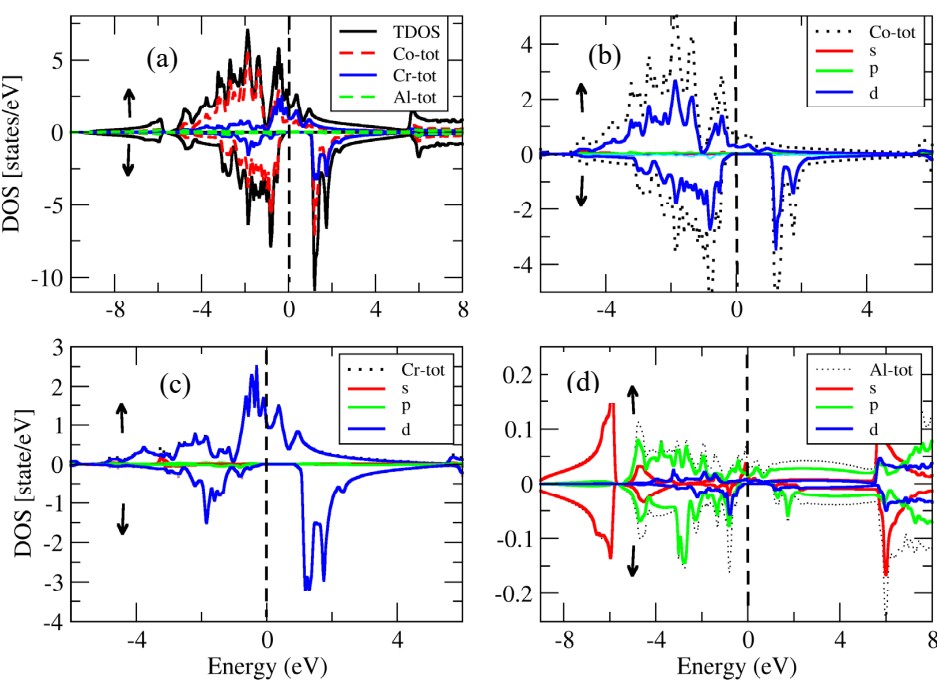

**Figure 15.** (**a**) Total density of states for the conventional Co$_2$CrAl compound and the partial density of states for (**b**) Co atom, (**c**) Cr atom and (**d**) Al atom.

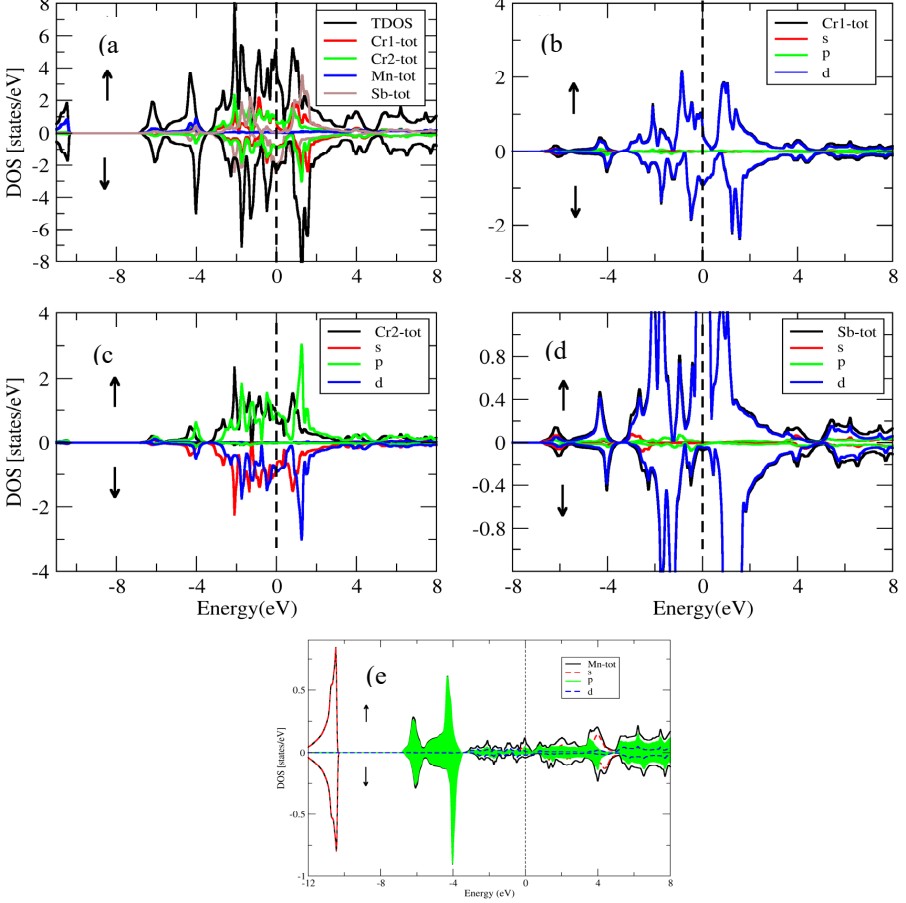

**Figure 16.** (**a**) Total density of states for the inverse Cr$_2$MnSb compound and partial density of states for (**b**) Cr1 atom, (**c**) Cr2 atom, (**d**) Sb atom and (**e**) Mn atom.

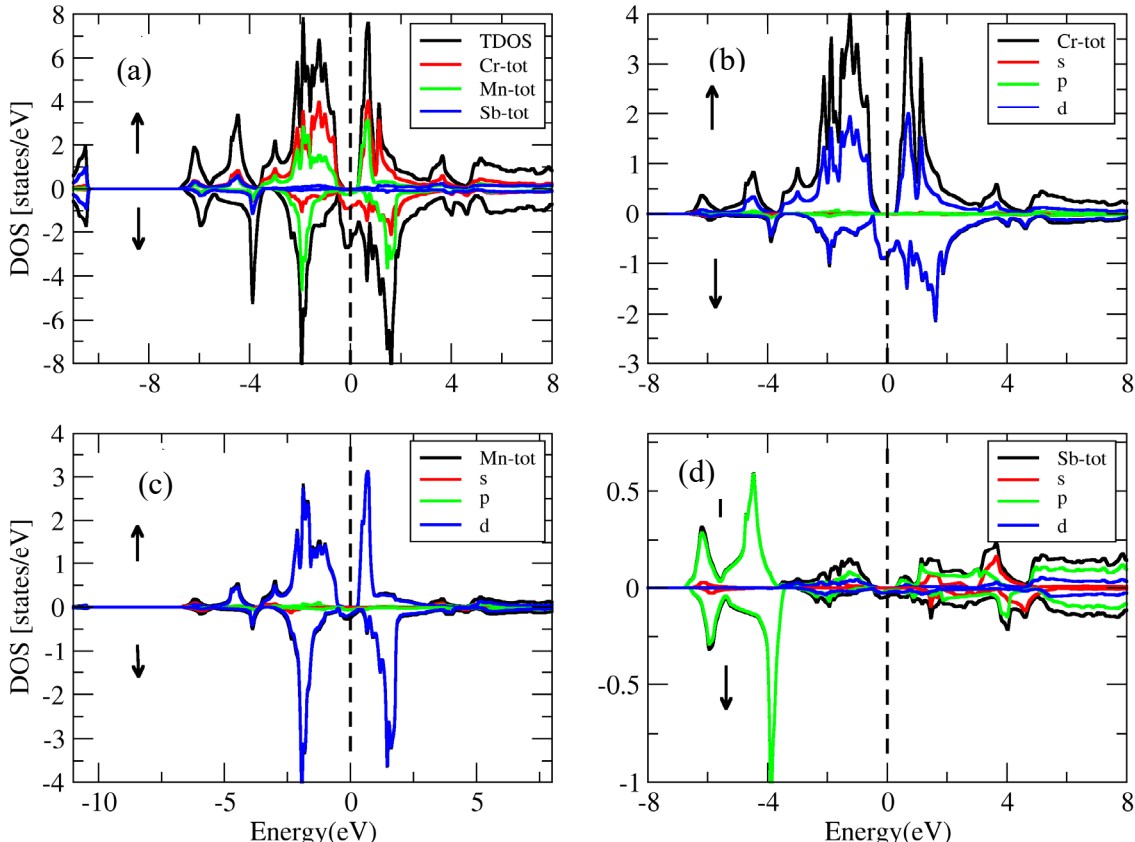

**Figure 17.** (**a**) Total density of states for the conventional $Cr_2MnSb$ compound and the partial density of states for (**b**) Cr atom, (**c**) Mn atom and (**d**) Sb atom.

In the spin-up sector of the inverse $Cr_2MnSb$, (Figure 16), the valence band was attributed to the d-state of Cr, the d-state of Mn, the s-state and the p-state of Sb. On the other hand, the conduction band resulted from the d-state of Cr, the d-state of Mn, the minor effect of the s-state and Sb's p-state. In the spin-down channel of the inverse $Cr_2MnSb$, the valence band was attributed to the minor contribution of the d-state of Cr, the d-state of Mn, the s-state and the p-state of Sb. Meanwhile, the conduction band was attributed to the d-state of Cr, the d-state of Mn, the minor contribution of the s-state and the p-state of Sb.

In the conventional $Cr_2MnSb$ (Figure 17) spin-up sector, the valence band was attributed to the d-state of Cr, the d-state of Mn, the s-state and the p-state of Sb. On the other hand, the conduction band was attributed to the d-state of Cr, the minor effect of the s-state and the p-state of Sb. In the spin-down channel of the conventional $Cr_2MnSb$, the valence band was attributed to the d-state of Cr, the d-state of Mn, the s-state and the p-state of Sb, while the conduction band was the result of the d-state of Cr, the minor effect of the s-state and the p-state of Sb.

### 3.4. Elastic Properties

This part presents the computation of the bulk modulus (*B*), the shear modulus (*S*), the elastic constants ($C_{ij}$), the $B/S$ ratio, Poisson's ratio, Young's modulus (*Y*) and the anisotropic factor (**A**) of the inverse and conventional Heusler $Cr_2MnSb$ and $Co_2CrAl$ compounds. The standard mechanical stability condition or cubic crystal [32] was $C_{11} > 0$, $C_{11} + 2C_{12} > 0$, $C_{11} - C_{12} > 0$ and $C_{44} > 0$.

Table 8 presents our calculations for the inverse and conventional Heusler $Cr_2MnSb$ and $Co_2CrAl$ compounds. We concluded that the inverse and conventional Heusler $Co_2CrAl$ were mechanically stable. The inverse $Cr_2MnSb$ was found to be mechan-

ically unstable in the ferromagnetic state. On the other hand, the conventional $Cr_2MnSb$ was mechanically stable in the ferromagnetic state.

**Table 8.** Reuss's bulk modulus (*B*), shear modulus (*S*), elastic constants ($C_{ij}$), B/S ratio, Poisson's ratio (*v*), Young's modulus (*Y*) and anisotropic factor (*A*) of the FM conventional and inverse Heusler ($Co_2CrAl$, $Cr_2MnSb$) compounds.

| Materials | $C_{11}$ (GPa) | $C_{12}$ (GPa) | $C_{44}$ (GPa) | *B* (GPa) | *S* (GPa) | B/S | *Y* (GPa) | *v* | *A* |
|---|---|---|---|---|---|---|---|---|---|
| Conventional $Co_2CrAl$ | 268.592 | 169.462 | 154.343 | 202.505 | 83.628 | 2.422 | 220.528 | 0.319 | 3.114 |
| Inverse $Co_2CrAl$ | 275.6864 | 234.2043 | 138.9022 | 248.032 | 42.364 | 5.855 | 120.246 | 0.470 | 6.697 |
| Inverse $Cr_2MnSb$ | 197.1538 | 138.7903 | −488.5573 | 158.245 | 80.134 | 1.975 | 205.683 | 0.283 | −16.7 |
| Conventional $Cr_2MnSb$ | 267.3836 | 224.7424 | 100.9041 | 238.956 | 40.474 | 5.904 | 114.933 | 0.420 | 4.732 |

We used the Reuss approximation [33] to calculate the bulk and shear modulus. The following equation can be used to calculate the Reuss shear modulus $S_R$:

$$S_R = \frac{5C_{44}(C_{11} - C_{12})}{4C_{44} + 3(C_{11} - C_{12})} \tag{3}$$

The following equation gives the cubic structure's bulk modulus:

$$B = \frac{1}{3}(C_{11} + 2C_{12}) \tag{4}$$

The Young modulus (*Y*) is given by the following:

$$Y = \frac{9BS_R}{(S_R + 3B)} \tag{5}$$

The anisotropic factor and Poisson's ratio are given by the following:

$$A = \frac{2C_{44}}{C_{11} - C_{12}} \tag{6}$$

$$v = \frac{3B - 2S_R}{2(3B + S_R)} \tag{7}$$

Reuss's bulk modulus (*B*), the shear modulus (*S*), the elastic constants ($C_{ij}$), the B/S ratio, Poisson's ratio (*v*), Young's modulus (*Y*) and the anisotropic factor (*A*) of the FM conventional and inverse Heusler $Co_2CrAl$ and $Cr_2MnSb$ compounds are shown in Table 8.

A material's hardness is measured by its shear modulus and bulk modulus [33]. Therefore, the ratio *B*/*S* measures a specific material's brittleness and ductility. A material is ductile when *B*/*S* < 1.75. Otherwise, it is brittle [34]. From the present calculations in Table 8, the *B*/*S* ratios of the inverse and conventional Heusler $Co_2CrAl$ compounds were 5.855 and 2.422, respectively. Both the inverse and conventional Heusler compounds were ductile in nature, depending on the *B*/*S* ratio values. The *B*/*S* ratio values for the inverse and conventional Heusler $Co_2CrAl$, compounds were found to be 1.975 and 5.904, respectively. Both the inverse and conventional Heusler $Cr_2MnSb$ compounds had a ductile character, depending on the B/S ratio values.

The stiffness of materials is measured using Young's modulus. Materials with a higher Young's modulus (*Y*) value are stiffer. Poisson's ratio can be employed for understanding the character of bonding and stability of a material. A Poisson's ratio value higher than

0.26 indicates that the material is ductile; otherwise, it is brittle [35]. From the present calculations summarized in Table 8, the Poisson's ratio values of the inverse and conventional Heusler $Co_2CrAl$ compounds were 0.470 and 0.319, respectively. Depending on the Poisson's ratio values, both inverse and conventional Heusler compounds had a ductile nature. The Poisson's ratio values for the inverse and conventional Heusler $Cr_2MnSb$ compounds were found to be 0.283 and 0.420, respectively. Depending on the Poisson's ratio values, both the inverse and conventional Heusler $Cr_2MnSb$ compounds had a ductile nature. Poisson's ratio for compounds with covalent bonds is lower than 0.25, while for compounds with dominating ionic bonds, Poisson's ratio lies between 0.25 to 0.50. From Table 8, it appeared that the conventional and inverse Heusler $Co_2CrAl$ and $Cr_2MnSb$ compounds had prominent ionic bonds. In the same vein, elastic anisotropy is a crucial parameter for measuring the level of material's anisotropy [36]. The value of A is unity for an isotropic material. Otherwise, the elastic anisotropy of the material is elastic [37]. The present values of the anisotropy factor in Table 8 for the inverse and conventional Heusler $Co_2CrAl$ and $Cr_2MnSb$ compounds showed that these compounds were anisotropic elasticity.

## 4. Conclusions

This study focused on the elastic, magnetic, electronic and structural properties of inverse and conventional Heusler ($Co_2CrAl$, $Cr_2MnSb$) compounds. The results showed that the conventional Heusler $Co_2CrAl$, the conventional and inverse Heusler $Cr_2MnSb$ and the tetragonal (139) Heusler $Co_2CrAl$ and $Cr_2MnSb$ compounds were half-metals. This half-metallic character is a promising characteristic of materials for spintronic applications. The indirect energy gap of the conventional Heusler $Co_2CrAl$ compound was 0.6 eV within the PBE-GGA scheme. The energy band gap within the mBJ-GGA scheme for the conventional Heusler $Co_2CrAl$ compound was computed to be 0.9 eV. Within the PBE-GGA technique, the inverse and conventional Heusler $Cr_2MnSb$ compounds had a direct energy band gap of 0.8 eV and 0.9 eV, respectively. Within the mBJ-GGA method, the energy gaps for the inverse and conventional Heusler $Cr_2MnSb$ compounds were 0.9 eV and 1 eV, respectively. The tetragonal Heusler $Co_2CrAl$ and $Cr_2MnSb$ compounds had direct energy band gaps and were 0.8 eV and 0.9 eV, respectively, within the PBE-GGA method. We discovered that the conventional Heusler $Co_2CrAl$ compound was a ferromagnetic compound with a total magnetic moment of $M^{tot} = 3$ $\mu_B$. On the other hand, the total magnetic moment for the inverse $Co_2CrAl$ compound was $M^{tot} = 0.831$ $\mu_B$. The conventional and tetragonal Heusler $Cr_2MnSb$ compounds had a small total magnetic moment, which meant that these compounds were ferromagnetic. We found that the conventional and inverse Heusler $Co_2CrAl$ compounds were mechanically stable. However, the inverse $Cr_2MnSb$ compound was mechanically unstable in the ferromagnetic state. On the other hand, the conventional $Cr_2MnSb$ is mechanically stable in the ferromagnetic state. The B/S values indicated that the inverse and conventional Heusler compounds for both $Cr_2MnSb$ and $Co_2CrAl$ had ductile characteristics. From the Poisson's ratio values, we found that the conventional and inverse Heusler $Co_2CrAl$ and $Cr_2MnSb$ compounds had dominant ionic bonds. Finally, the conventional and inverse Heusler $Co_2CrAl$ and $Cr_2MnSb$ compounds were anisotropy elastic.

**Author Contributions:** M.S.A.-J.: conceptualization, methodology, software, investigation, validation, visualization, formal analysis, writing—review & editing, supervision, project administration. S.J.Y.: data curation, methodology, formal analysis, writing—original draft, software. S.A.A.: data curation, formal analysis, writing—review & editing, software, investigation, validation. A.A.M.: data curation, methodology, formal analysis, writing—review & editing, software. D.A.-B.: data curation, methodology, formal analysis, software. M.F.: data curation, methodology, formal analysis, writing—review & editing, software. R.K.: methodology, formal analysis, writing—review & editing. All authors have read and agreed to the published version of the manuscript.

**Funding:** This research received no external funding.

**Data Availability Statement:** The data that support the findings of this study are available from the corresponding author upon reasonable request.

**Conflicts of Interest:** The authors declare that they have no known competing financial interests or personal relationships that could have appeared to influence the work reported in this paper.

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
