# Peer review of "The Structural, Electronic, Magnetic and Elastic Properties of Full-Heusler Co2CrAl and Cr2MnSb: An Ab Initio Study"

_crystals, doi:10.3390/cryst12111580_

Round 1
Reviewer 1 Report
The paper looks to me very well written and presented, with some improvements light the "CR" element in the abstract, ex. CR2CrAl, which I don't get, and the lines in the figure which be made bolder.
Apart these, relevant point that must be addressed before the publications are:
1) provide a demonstrate that the 10*10*10 sampling k-mesh is fine; a reader might think, as I did, that it is too coarse. So the authors must demonstrate it is okay to achieve reasonable results.
2) compare the obtained results with previous calculations (if any) and with the available experimental data.
Author Response
Dear Professor,
Thank you for your fruitful and helpful comments and suggestions sent to us for improving our manuscript. The English language of the manuscript has been thoroughly revised typographical and grammatical by an English native speaker. We have revised the manuscript accordingly, and the detailed corrections are given below and they are highlighted in red color in the text of the revised manuscript.
Point 1: provide a demonstrate that the 10*10*10 sampling k-mesh is fine; a reader might think, as I did, that it is too coarse. So, the authors must demonstrate it is okay to achieve reasonable results.
Response 1: A sufficiently dense k-point grid mesh sampling was used to achieve a convergence of the total energy, Etotal . The self-consistent calculations of Etotal of the unit cell are converged when the Etotal is stable within 10-5 Ry. We have tried several sampling k-mesh before getting 10 x 10 x10 mesh of points until getting the minimum total energy with excellent convergent. For example, if we keep increasing the k-mesh for more than 10 x 10 x 10 it will not change the value of the total energy rather than time consuming of the calculations. Somewhere we have to stop the iterations on a certain k-mesh sampling.
Point 2: compare the obtained results with previous calculations (if any) and with the available experimental data.
Response 2: Our obtained results are compared with other experimental results (References 19 & 20) and with other theoretical results (References 21 & 22). Please see Tables 1, 2, 3 and 4.

Reviewer 2 Report
Sara et al. studied the electronic, magnetic and elastic properties of normal and Inverse Heusler alloys by using density functional theory. The manuscript is well written but authors need to revise it before publication in the Crystal.
The comments are given below:
1) Please improve the quality of figures, specially the crystal structures of these alloys.
2) Check the suffix of the compounds, replace Cr2CrAl with Cr2CrAl
3) Show spin up and spin down in same figures.
4) Add one paragraph of experimental work related to these alloys in the introduction section.
5) Replace C11 by C11 and similarly for other elastic constants.
6) Why the total magnetic moments of normal and inverse alloys shows huge different? please explain it.
7) Cite the following articles which are related this work:
i) https://doi.org/10.1016/j.vacuum.2019.108931
ii)https://doi.org/10.1016/j.vacuum.2019.03.049
8) You have used two exchange-correlation functional such as GGA and mBJ-GGA. Please also write about these functional in the computational method.
Author Response
Response to Reviewer 2 Comments
Dear Professor,
Thank you for your fruitful and helpful comments and suggestions sent to us for improving our manuscript. The English language of the manuscript has been thoroughly revised typographical and grammatical by an English native speaker. We have revised the manuscript accordingly, and the detailed corrections are given below and they are highlighted in red color in the text of the revised manuscript.
Point 1: Please improve the quality of figures, specially the crystal structures of these alloys.
Response 1: Done.
Point 2: Check the suffix of the compounds, replace Cr2CrAl with Cr2CrAl
Response 2: Done.
Point 3:. Show spin up and spin down in same figures
Response 3: Done.
Point 4: Add one paragraph of experimental work related to these alloys in the introduction section.
Response 4: One paragraph on the experimental work has been added in the introduction section and References 25 & 26 have been cited in our work. References 19 & 20 related to the experimental results have been already cited into our work.
Point 5: Replace C11 by C11 and similarly for other elastic constants.
Response 5: Done.
Point 6: Why the total magnetic moments of normal and inverse alloys show huge different? please explain it.
Response 6: Done. A paragraph has been added into the magnetic properties section. The physics interpretation behind this huge difference between the total spin magnetic moment of normal and inverse Co2CrAl is due to the antiparallel exchange interaction between the Cr atom and Co atom in the case of inverse Co2CrAl whereas it is direct interaction in the case of the normal phase.
Point 7:. Cite the following articles which are related this work:
- i) https://doi.org/10.1016/j.vacuum.2019.108931
ii)https://doi.org/10.1016/j.vacuum.2019.03.049
Response 7: Done. The above two papers have been cited at the end of the introduction section.
Point 8: You have used two exchange-correlation functional such as GGA and mBJ-GGA. Please also write about these functional in the computational method.
Response 8: Done. Generalized Gradient Approximation (GGA) depends on the local gradient of the electronic density in addition to the value of the density, giving a more accuarte description of variations in the electron-electron interactions. GGA functionals provide severe underestimation of the energy band gaps. Modified Becke-Johnson (mBJ-GGA) functional is introduced to improve the energy band gaps by fixing the problem of underestimation of the energy band gaps to be comparable to the experimental results. Therefore, the mBJ-GGA) is an efficient tool to be used to overcome the well-known underestimation of the band gap values by the GGA method.

Reviewer 3 Report
This manuscript summarizes electronic, magnetic and mechanical properties of Co2CrAl and Cr2MnSb. This study can be published, after authors thoroughly revise the English of the manuscript. I have one specific question: What are the key findings in this study? Author should also improve the "introduction" significantly.
I also mentioned few typos. There might be more in the manuscript.
11. Abstract: Replace “CR2CrAl” by “Co2CrAl”; “CR2MnSb” by “Cr2MnSb”
22. Replace “Normal” Heusler alloy by “Conventional” Heusler throughout the manuscript
33. Introduction: Replace “Heusler compounds have half-metallic (HM) character.” by “Some of the Heusler compounds have half-metallic (HM) character.” Include some references.
(Not all Heusler alloys are half-metallic)
44. Replace “Co2MnGE” by “Co2MnGe”
55. Page 2: line 65-line 72: Please rewrite this part. The meaning is not clear.
Starting from “Atsufumi Hirohata et al, focused …. AFM behavior higher than room temperature”.
6. Replace “Plain waves” by “Plane-wave” (page2, line 97)
77. Modify : “the normal Heusler Cr2MnSb is a ferrimagnetism…” (page 7, line 177)
88. Page 14, line 265, what is Ce2MnSb?
99. Whys so many figures? You may reduce the number of figures by presenting spin-up and spin-down dos in a same figure.
Author Response
Response to Reviewer 3 Comments
Dear Professor,
Thank you for your fruitful and helpful comments and suggestions sent to us for improving our manuscript. The English language of the manuscript has been thoroughly revised typographical and grammatical by an English native speaker. We have revised the manuscript accordingly, and the detailed corrections are given below and they are highlighted in red color in the text of the revised manuscript. The introduction has been improved significantly by adding a new four references. The key findings in this study can be summarized as follows: 1- Co2CrAl is found to have a half-metallic character. 2- Normal and inverse full-Heusler Co2CrAl are mechanically stable in the ferromagnetic state
3- Normal Heusler is mechanically stable in the ferromagnetic state, while inverse is not.
4- Normal and tetragonal Heusler are ferrimagnetic due to their small total magnetic moment.
5- Normal and inverse Heusler for both and have ductile natures.
6- Normal and inverse Heusler, are found to be elastically anisotropic.
The HM Heusler alloys are good candidates for the application of spintronic devices as these alloys have a 100% spin polarization ratio of conductive electrons, compatible lattice structure to the widely used semiconductors and the other excellent properties, i.e., high curie temperature and ease fabrication.
Point 1: Abstract: Replace “CR2CrAl” by “Co2CrAl”; “CR2MnSb” by “Cr2MnSb”
Response 1: Done.
Point 2: Replace “Normal” Heusler alloy by “Conventional” Heusler throughout the manuscript
Response 2: Done. Normal Heusler has been replaced by Conventional through the manuscript.
Point 3:. Introduction: Replace “Heusler compounds have half-metallic (HM) character.” by “Some of the Heusler compounds have half-metallic (HM) character.” Include some references.
(Not all Heusler alloys are half-metallic)
Response 3: Done. The references related to the some of the half-metallic Heusler compounds have been included.
Point 4: Replace “Co2MnGE” by “Co2MnGe”
Response 4: Done.
Point 5: Page 2: line 65-line 72: Please rewrite this part. The meaning is not clear.
Starting from “Atsufumi Hirohata et al, focused …. AFM behavior higher than room temperature”..
Response 5: Done. The part has been rewritten as following: “Atsufumi Hirohata et al. [23] reviewed the development of anti-ferromagnetic (AFM) Heusler alloys for the replacement of iridium as a critical raw material (CRMs). They have established correlations between the crystalline structure of these alloys and the magnetic properties, i.e., antiferromagnetism. This study reveals that the Heusler alloys consisting of elements with moderate magnetic moments require perfectly or partially ordered crystalline structures to exhibit AFM behavior. Using elements with large magnetic moments, a fully disordered structure is found to show either AFM or ferrimagnetic (FIM) behavior. The considered alloys may become useful for device applications by additional increase of their anisotropy and grain volume to sustain the AFM behavior above room temperature”.
Point 6: Replace “Plain waves” by “Plane-wave” (page2, line 97)
Response 6: Done.
Point 7:. Modify : “the normal Heusler Cr2MnSb is a ferrimagnetism…” (page 7, line 177)
Response 7: Done. The statement has been modified as following: “We found that the normal Heusler Cr2MnSb has a small total magnetic moment (non-zero total magnetization) due to the decrease of the atomic disorder in the Mn-Sb sublattice. This implies that the normal Heusler Cr2MnSb has a ferrimagnetic order”.
Point 8: Page 14, line 265, what is Ce2MnSb?
Response 8: Done.
Point 9: Whys so many figures? You may reduce the number of figures by presenting spin-up and spin-down dos in a same figure.
Response 9: Done. The figures have been reduced.

Round 2
Reviewer 2 Report
There are many paragraphs in the introduction section .
please merge these paragraphs and make four paragraphs
Reviewer 3 Report
Authors have taken care of my comments. The manuscript can now be published.